# Stochastic Kernel Regularisation Improves Generalisation in Deep Kernel Machines

**Edward Milsom**
School of Mathematics
University of Bristol
edward.milsom@bristol.ac.uk

**Ben Anson**
School of Mathematics
University of Bristol
ben.anson@bristol.ac.uk

**Laurence Aitchison**
School of Engineering Mathematics and Technology
University of Bristol
laurence.aitchison@gmail.com

## Abstract

Recent work developed convolutional deep kernel machines, achieving 92.7% test accuracy on CIFAR-10 using a ResNet-inspired architecture, which is SOTA for kernel methods. However, this still lags behind neural networks, which easily achieve over 94% test accuracy with similar architectures. In this work we introduce several modifications to improve the convolutional deep kernel machine's generalisation, including stochastic kernel regularisation, which adds noise to the learned Gram matrices during training. The resulting model achieves 94.5% test accuracy on CIFAR-10. This finding has important theoretical and practical implications, as it demonstrates that the ability to perform well on complex tasks like image classification is not unique to neural networks. Instead, other approaches including deep kernel methods can achieve excellent performance on such tasks, as long as they have the capacity to learn representations from data.

## 1 Introduction

Neural network Gaussian Processes (NNGPs) (Lee et al., 2017) are a key theoretical tool in the study of neural networks. When a randomly initialised neural network is taken in the infinite-width limit, it becomes equivalent to a Gaussian process with the NNGP kernel. A large body of work has focused on improving the predictive accuracy of NNGPs (Novak et al., 2018; Garriga-Alonso et al., 2018; Arora et al., 2019; Lee et al., 2020; Li et al., 2019; Shankar et al., 2020; Adlam et al., 2023), but they still fall short of finite neural networks (NNs). One hypothesis is that this is due to the absence of representation learning; the NNGP kernel is a fixed and deterministic function of its inputs (MacKay, 1998; Aitchison, 2020), but finite neural networks fit their top-layer representations to the task; this aspect of learning is considered vital for the success of contemporary deep learning (Bengio et al., 2013; LeCun et al., 2015). Exploring this hypothesis through the development of NNGP-like models equipped with representation learning could deepen our understanding of neural networks. Although there are some theoretical frameworks for representation learning (Dyer & Gur-Ari, 2019; Hanin & Nica, 2019; Aitchison, 2020; Li & Sompolinsky, 2020; Antognini, 2019; Yaida, 2020; Naveh et al., 2020; Zavatone-Veth et al., 2021; Zavatone-Veth & Pehlevan, 2021; Roberts et al., 2021; Naveh & Ringel, 2021; Halverson et al., 2021; Seroussi et al., 2023), they are generally not scalable enough to handle important deep learning datasets like CIFAR-10.

One recent promising proposal in this area is deep kernel machines (DKMs Yang et al., 2023; Milsom et al., 2024). DKMs are an entirely kernel-based method (i.e. there are no weights and

38th Conference on Neural Information Processing Systems (NeurIPS 2024).

no features) that is nonetheless able to learn representations from data. DKMs narrow the gap to DNNs, achieving 92.7% performance on CIFAR-10 Milsom et al., 2024 vs 91.2% (Adlam et al., 2023) for traditional infinite-width networks. However, 92.7% is still far from standard DNNs like ResNets, which achieve around 95% performance on CIFAR-10. Here, we narrow this gap further, achieving 94.5% performance on CIFAR-10, by introducing two key modifications. First, we introduce a regularisation scheme inspired by dropout (Srivastava et al., 2014) where we randomly sample positive-definite matrices in place of our learned Gram matrices. We refer to this scheme as stochastic kernel regularisation (SKR). Second, we use single-precision floating point arithmetic to greatly accelerate training, allowing us to train for more epochs under a fixed computational budget. This presents challenges concerning numerical stability, and we develop a number of mitigations to address these.

## 2 Background: Convolutional Deep Kernel Machines

Here, we provide a brief overview of convolutional deep kernel machines. For a more in-depth introduction see Appendix A, or Milsom et al. (2024). Deep kernel machines are a class of supervised learning algorithms that compute a kernel from inputs and transform this kernel through a series of learnable mappings over Gram matrices. We can then perform prediction with the top layer kernel representation using e.g. GP regression/classification. A deep kernel machine is similar in structure to a deep Gaussian process, with the key difference being that instead of features $\mathbf{F}^\ell \in \mathbb{R}^{P \times N_\ell}$ at each layer, where $P$ is the number of datapoints and $N_\ell$ is the number of features at layer $\ell$, we work with Gram matrices $\mathbf{G}^\ell \in \mathbb{R}^{P \times P}$. We define the Gram matrices as the (normalised) dot product between features:

$$\mathbf{G}^\ell = \frac{1}{N_\ell} \mathbf{F}^\ell (\mathbf{F}^\ell)^T. \tag{1}$$

Many common kernel functions $\mathbf{K}(\cdot)$, such as the arccos (Cho & Saul, 2009) and squared-exponential, depend on features only through their pairwise dot products, and thus can be computed in this reparametrised model as $\mathbf{K}(\mathbf{G}^\ell)$ instead of the usual $\mathbf{K}(\mathbf{F}^\ell)$. To obtain a deep kernel machine, all layers are taken in the infinite-width limit, i.e. $N_\ell \to \infty$, and the likelihood function is scaled to retain representation learning. Under this limit the Gram matrices, $\{\mathbf{G}^\ell\}_{\ell=1}^L$, which were previously random variables whose distribution we might approximate through variational inference (see "deep kernel processes" Aitchison et al., 2021; Ober & Aitchison, 2021; Ober et al., 2023), become deterministic / point-distributed, and can therefore be treated as learnable parameters. We learn the Gram matrices by optimising the deep kernel machine objective (which is itself derived from the evidence lower-bound (ELBO) for variational inference in the infinite-width limit) using gradient ascent:

$$\mathcal{L}(\mathbf{G}^1, \ldots, \mathbf{G}^L) = \log \mathrm{P}\left(\mathbf{Y}|\mathbf{G}^L\right) - \sum_{\ell=1}^L \nu_\ell \, \mathrm{D}_{\mathrm{KL}}\left(\mathcal{N}\left(\mathbf{0}, \mathbf{G}^\ell\right)\|\mathcal{N}\left(\mathbf{0}, \mathbf{K}(\mathbf{G}^{\ell-1})\right)\right). \tag{2}$$

Since the computations involved are very similar to Gaussian processes, DKMs naturally scale like $\mathcal{O}(P^3)$ with the number of training points. Implementations of shallow Gaussian processes often compute the full kernel matrix using lazy evaluation (Novak et al., 2019; Gardner et al., 2018; Charlier et al., 2021) which avoids storing the entire kernel matrix in memory at once. This process is often very slow (Google's neural tangents library takes 508 GPU hours to compute the Myrtle-10 kernel on CIFAR-10 Google, 2020; Novak et al., 2019), and therefore infeasible for our setting where the model is both deep, and requires thousands of iterations of gradient ascent. Instead, previous work on deep kernel machines utilises sparse variational inducing point schemes, inspired by similar work on deep Gaussian processes (Salimbeni & Deisenroth, 2017). These schemes replace the $P_t$ training points with $P_i$ pseudo-datapoints, where $P_i \ll P_t$, which leads to $\mathcal{O}(P_i^3 + P_i^2 P_t)$ computations which are much cheaper. In deep Gaussian processes, a separate set of $P_i^\ell$ inducing points is learned for each layer $\ell$, approximating the input-output functions locally. Deep kernel machines typically use an analogous inducing point scheme, learning an inducing Gram matrix $\mathbf{G}_{ii}^\ell \in \mathbb{R}^{P_i^\ell \times P_i^\ell}$ at each layer (rather than the full Gram matrix for all training points, $\mathbf{G}^\ell \in \mathbb{R}^{P_t \times P_t}$) by optimising a similar objective to Eq. (2) (see Eq. 27 in Appendix A for further details). Prediction of train/test points is described in Algorithm 1, but at a high level, it is similar to Gaussian process prediction, except instead of working with a feature vector partitioned into inducing points and train/test points, we have a Gram matrix partitioned into 4 blocks,

$$\frac{1}{N_\ell}\begin{pmatrix}\mathbf{F}_i^\ell\\\mathbf{F}_t^\ell\end{pmatrix}\begin{pmatrix}\mathbf{F}_i^\ell\\\mathbf{F}_t^\ell\end{pmatrix}^T = \begin{pmatrix}\frac{1}{N_\ell}\mathbf{F}_i^\ell(\mathbf{F}_i^\ell)^T & \frac{1}{N_\ell}\mathbf{F}_i^\ell(\mathbf{F}_t^\ell)^T\\\frac{1}{N_\ell}\mathbf{F}_t^\ell(\mathbf{F}_i^\ell)^T & \frac{1}{N_\ell}\mathbf{F}_t^\ell(\mathbf{F}_t^\ell)^T\end{pmatrix} = \begin{pmatrix}\mathbf{G}_{ii}^\ell & (\mathbf{G}_{ti}^\ell)^T\\\mathbf{G}_{ti}^\ell & \mathbf{G}_{tt}^\ell\end{pmatrix}. \tag{3}$$

The goal in prediction is to compute the conditional expectation of $\mathbf{G}_{\text{ti}}^{\ell} \in \mathbb{R}^{P_{\text{t}}^{\ell} \times P_{\text{i}}^{\ell}}$ and $\mathbf{G}_{\text{tt}}^{\ell} \in \mathbb{R}^{P_{\text{t}}^{\ell} \times P_{\text{t}}^{\ell}}$ conditioned on the learned inducing block $\mathbf{G}_{\text{ii}}^{\ell} \in \mathbb{R}^{P_{\text{i}}^{\ell} \times P_{\text{i}}^{\ell}}$.

Convolutional deep kernel machines combine deep kernel machines with convolutional kernel functions from the infinite-width neural network literature / Gaussian process literature (van der Wilk et al., 2017; Dutordoir et al., 2020; Garriga-Alonso et al., 2018; Novak et al., 2018). In these settings, kernel matrices are of size $PWH \times PWH$ where $P$ is the number of training images, and $W, H$ are the width and height of the images, respectively. In particular, this implies that inducing Gram matrices $\mathbf{G}_{\text{ii}}^{\ell}$ are of size $P_{\text{i}}^{\ell}WH \times P_{\text{i}}^{\ell}WH$ which is too expensive even for CIFAR-10 (e.g. 10 $32 \times 32$ inducing CIFAR-10 images results in a $10,240 \times 10,240$ kernel matrix). To avoid this exorbitant computational cost, Milsom et al. (2024) proposed an inducing point scheme where the learned inducing blocks $\mathbf{G}_{\text{ii}}^{\ell}$ have no "spatial" dimensions, i.e. they are of size $P_{\text{i}}^{\ell} \times P_{\text{i}}^{\ell}$, whilst the predicted train / test blocks $\mathbf{G}_{\text{ti}}^{\ell} \in \mathbb{R}^{P_{\text{t}}WH \times P_{\text{i}}^{\ell}}, \mathbf{G}_{\text{it}}^{\ell} \in \mathbb{R}^{P_{\text{i}}^{\ell} \times P_{\text{t}}WH}, \mathbf{G}_{\text{tt}}^{\ell} \in \mathbb{R}^{P_{\text{t}}WH \times P_{\text{t}}WH}$ retain their spatial dimensions. They use linear maps $\mathbf{C}^{\ell} \in \mathbb{R}^{DP_{\text{i}}^{\ell} \times P_{\text{i}}^{\ell-1}}$ to map the $P_{\text{i}}^{\ell-1}$ non-spatial inducing points into $P_{\text{i}}^{\ell}$ patches with $D$ pixels, which can be used by the convolutional kernel.

The full prediction algorithm is given in Algorithm 1. In practice, the IID likelihood functions used at the output layer mean only the diagonal of $\mathbf{G}_{\text{tt}}^{\ell}$ is needed, dramatically reducing the computation and storage requirements to a vector $\mathbf{G}_{\text{t}}^{\ell} := \text{diag}(\mathbf{G}_{\text{tt}}^{\ell}) \in \mathbb{R}^{P_{\text{t}}^{\ell}WH}$, and only minibatches of the data are required (Yang et al., 2023). The parameters are updated via gradient descent by computing the objective (Equation 27 in Appendix A) using the predictions, and backpropagating.

# 3  Methods

We seek to improve the generalisation of the convolutional deep kernel machine via two main strategies. First, we seek to reduce overfitting of representations by introducing randomness to the learned inducing Gram matrices at train time, replacing them with samples from the Wishart distribution. We refer to this process as "stochastic kernel regularisation" to avoid ambiguity with terms like "random sampling". Second, we improve the numerical stability of the algorithm enough to utilise lower-precision TF32 cores in modern NVIDIA GPUs, which are significantly faster but more prone to round-off errors. Using TF32 cores makes training roughly $5\times$ faster than the implementation in Milsom et al. (2024), which used double-precision floating points, and therefore allows us to train for significantly more epochs. Numerical stability is improved through a combination of stochastic kernel regularisation and a Taylor approximation to the problematic log-determinant term in the objective function, which we show has no negative effect on predictive performance in our ablation experiments. We also observed that keeping the regularisation strength $\nu$ in the DKM objective (Equation 2) non-zero was crucial in preserving numerical stability.

## 3.1  Stochastic kernel regularisation

Milsom et al. (2024) observed that convolutional deep kernel machines suffer from overfitting. Under the infinite-width limit, the distributions over Gram matrices become point distributions, and offer no stochastic regularising effect. Inspired by dropout in neural networks (Srivastava et al., 2014), we introduce random noise into the training process to reduce overfitting of representations. Specifically, we replace the inducing Gram matrices $\mathbf{G}_{\text{ii}}^{\ell}$ at each layer with a sample $\tilde{\mathbf{G}}_{\text{ii}}^{\ell}$ from the Wishart distribution,

$$\tilde{\mathbf{G}}_{\text{ii}}^{\ell} \sim \mathcal{W}(\mathbf{G}_{\text{ii}}^{\ell}/\gamma, \gamma), \tag{4}$$

which has expectation $\mathbf{G}_{\text{ii}}^{\ell}$ and variance inversely proportional to $\gamma$. Strictly speaking, the Wishart distribution has support over positive-definite matrices only. This positive-definiteness constraint corresponds to the requirement $\gamma \geq P_{\text{i}}^{\ell}$, which in turn upper-bounds the variance of the samples. In highly expressive models with many inducing points, it may be beneficial to have much higher variance samples than this, so we relax the constraint on $\gamma$, leading to potentially singular matrices, and then apply jitter to the samples, i.e. $\tilde{\mathbf{G}}_{\text{ii}}^{\ell} \mapsto \tilde{\mathbf{G}}_{\text{ii}}^{\ell} + \lambda\mathbf{I}$, to ensure positive-definiteness. Random sampling is disabled at test-time, though we still add the same jitter to eliminate bias between train and test predictions. We refer to this process as stochastic kernel regularisation (SKR).

**Algorithm 1** Convolutional deep kernel machine prediction. Changes from this paper are in red.

---

**Input:** Batch of datapoint inputs: $\mathbf{X}_t \in \mathbb{R}^{P_t W H \times \nu_0}$
**Output:** Distribution over predictions $\mathbf{Y}_t^*$
**Parameters:** Inducing inputs $\mathbf{X}_i$, inducing Gram matrices $\{\mathbf{G}_{ii}^\ell\}_{\ell=1}^L$, inducing output GP approximate posterior parameters $\boldsymbol{\mu}_1, \ldots, \boldsymbol{\mu}_{\nu_{L+1}}, \boldsymbol{\Sigma}$ (shared across classes), inducing "mix-up" parameters $\{\mathbf{C}^\ell\}_{\ell=1}^L$, where $\mathbf{C}^\ell \in \mathbb{R}^{DP_i^\ell \times P_i^{\ell-1}}$

Initialise full Gram matrix
$$\begin{pmatrix} \mathbf{G}_{ii}^0 & \mathbf{G}_{it}^0 \\ \mathbf{G}_{ti}^0 & \mathbf{G}_{tt}^0 \end{pmatrix} = \frac{1}{\nu_0} \begin{pmatrix} \mathbf{X}_i \mathbf{X}_i^T & \mathbf{X}_i \mathbf{X}_t^T \\ \mathbf{X}_t \mathbf{X}_i^T & \mathbf{X}_t \mathbf{X}_t^T \end{pmatrix}$$
**for** $\ell$ in $(1, \ldots, L)$ **do**

Apply kernel non-linearity $\boldsymbol{\Phi}(\cdot)$ (e.g. arccos kernel)
$$\begin{pmatrix} \boldsymbol{\Phi}_{ii} & \boldsymbol{\Phi}_{ti}^T \\ \boldsymbol{\Phi}_{ti} & \boldsymbol{\Phi}_{tt} \end{pmatrix} = \boldsymbol{\Phi}\left( \begin{pmatrix} \mathbf{G}_{ii}^{\ell-1} & (\mathbf{G}_{ti}^{\ell-1})^T \\ \mathbf{G}_{ti}^{\ell-1} & \mathbf{G}_{tt}^{\ell-1} \end{pmatrix} \right)$$
Apply convolution and "mix up" inducing points (see Appendix A or Milsom et al. (2024)). Indexing $d$ represents pixels within a patch, and $(i, r)$ denotes "image / feature map $i$, spatial location $r$"
$\mathbf{K}_{ii} = \frac{1}{D} \sum_d \mathbf{C}_d^\ell \boldsymbol{\Phi}_{ii} (\mathbf{C}_d^\ell)^T$
$\mathbf{K}_{ti(i,r)} = \frac{1}{D} \sum_d \boldsymbol{\Phi}_{ti(i,r+d)} (\mathbf{C}_d^\ell)^T$ i.e. $\mathbf{K}_{ti} = \text{conv2D}(\boldsymbol{\Phi}_{ti}, \text{filters} = \mathbf{C}^\ell)$
$\mathbf{K}_{tt(i,r),(j,s)} = \frac{1}{D} \sum_d \boldsymbol{\Phi}_{(i,r+d),(j,s+d)}^{tt}$ similar to the avg_pool2D operation
Apply stochastic kernel regularisation to inducing Gram matrix
$\tilde{\mathbf{G}}_{ii}^\ell \sim \mathcal{W}(\mathbf{G}_{ii}^\ell / \gamma, \gamma) + \lambda \mathbf{I}$
Predict train/test components of Gram matrix conditioned on inducing component $\tilde{\mathbf{G}}_{ii}^\ell$
$\mathbf{K}_{tt \cdot i} = \mathbf{K}_{tt} - \mathbf{K}_{ti} \mathbf{K}_{ii}^{-1} \mathbf{K}_{ti}^T$.
$\mathbf{G}_{ti}^\ell = \mathbf{K}_{ti} \mathbf{K}_{ii}^{-1} \tilde{\mathbf{G}}_{ii}^\ell$
$\mathbf{G}_{tt}^\ell = \mathbf{K}_{ti} \mathbf{K}_{ii}^{-1} \tilde{\mathbf{G}}_{ii}^\ell \mathbf{K}_{ii}^{-1} \mathbf{K}_{ti}^T + \mathbf{K}_{tt \cdot i}$
**end for**

Average over spatial dimension, forming an additive GP (van der Wilk et al., 2017). $(r)$ and $(s)$ index spatial locations in a feature map, and $S$ is the total number of spatial locations.
$\mathbf{G}_{ii}^{\text{Flat}} = \tilde{\mathbf{G}}_{ii}^L$
$\mathbf{G}_{ti}^{\text{Flat}} = \frac{1}{S} \sum_r \mathbf{G}_{ti(r)}^L$
$\mathbf{G}_{tt}^{\text{Flat}} = \frac{1}{S^2} \sum_{rs} \mathbf{G}_{tt(r),(s)}^L$
Final prediction using standard Gaussian process expressions
$$\begin{pmatrix} \mathbf{K}_{ii} & \mathbf{K}_{ti}^T \\ \mathbf{K}_{ti} & \mathbf{K}_{tt} \end{pmatrix} = \boldsymbol{\Phi}\left( \begin{pmatrix} \mathbf{G}_{ii}^{\text{Flat}} & (\mathbf{G}_{ti}^{\text{Flat}})^T \\ \mathbf{G}_{ti}^{\text{Flat}} & \mathbf{G}_{tt}^{\text{Flat}} \end{pmatrix} \right)$$
Sample features $\mathbf{f}_\lambda^{t;L+1}$ conditioned on $\mathbf{K}$ and inducing outputs $Q(\mathbf{f}_\lambda^{i;L+1}) \sim \mathcal{N}(\boldsymbol{\mu}_\lambda, \boldsymbol{\Sigma})$
$\mathbf{f}_\lambda^{t;L+1} \sim \mathcal{N}(\mathbf{K}_{ti} \mathbf{K}_{ii}^{-1} \boldsymbol{\mu}_\lambda, \mathbf{K}_{tt} - \mathbf{K}_{ti} \mathbf{K}_{ii}^{-1} \mathbf{K}_{it} + \mathbf{K}_{ti} \mathbf{K}_{ii}^{-1} \boldsymbol{\Sigma} \mathbf{K}_{ii}^{-1} \mathbf{K}_{it})$
Monte-Carlo average over softmax of features to obtain categorical distribution over classes
$\mathbf{Y}_t^* = \mathbb{E}\big[\text{softmax}(\mathbf{f}_1^{t;L+1}, \ldots, \mathbf{f}_{\nu_{L+1}}^{t;L+1})\big]$
Training: Compute DKM objective (Eq. 27) with Taylor approximation (Eq. 8) using true labels $\mathbf{Y}_t$ and backpropagate to update parameters.

---

## 3.2 Enabling lower-precision floating point arithmetic

Previous implementations of deep kernel machines (Yang et al., 2023; Milsom et al., 2024) used double-precision floating point arithmetic , which is very slow. Modern GPUs are highly optimised for lower-precision floating point operations. For example, the NVIDIA A100 GPU marketing material (NVIDIA, 2021) quotes 9.7 TFLOPS for FP64 operations, 19.5 TFLOPS for FP32 operations, and 156 TFLOPS for TensorFloat-32 (TF32) operations, a proprietary standard that has the 8-bit exponent (range) of FP32 but the 10-bit mantissa (precision) of FP16, for a total of 19 bits including the sign bit (Kharya, 2020). Therefore, switching from FP64 to TF32 numbers suggests potential speedups of up to $8\times$, though in reality speedups will be more modest as not all operations support TF32. Working with kernel methods in low precision arithmetic requires care to ensure numerical stability. For example, direct inversion of the kernel matrix $\mathbf{K}_{ii}$ should be avoided, and instead all operations

of the form $\mathbf{K}_{ii}^{-1}\mathbf{B}$ should instead be computed by factorising $\mathbf{K}_{ii}$ and solving the system $\mathbf{K}_{ii}\mathbf{X} = \mathbf{B}$ (Trefethen & Bau, 1997).

Unfortunately, the convolutional deep kernel machine as presented in Milsom et al. (2024) is highly unstable with low-precision arithmetic. Subroutines for the exact computation of Cholesky decompositions fail completely (halting any further progress in training) when large round-off errors accumulate. This problem is particularly acute when dealing with large kernel matrices, which are typically very ill-conditioned. The usual solution is to add jitter to the kernel matrices, but we found this was insufficient when using such low-precision arithmetic (Table 3). Instead, we hypothesised that the problem lay not in the kernel matrices $\mathbf{K}$, but rather in the learned inducing Gram matrices $\mathbf{G}_{ii}^{\ell}$. In particular, we observed that the condition number of $\mathbf{G}_{ii}^{\ell}$ tended to worsen over time (Fig. 1), suggesting that learning highly expressive representations led to ill-conditioned Gram matrices.

Though the stochastic kernel regularisation scheme we proposed did result in improved condition numbers during training (Fig. 1), we still observed occasional failures in our large-scale experiments on CIFAR-10 (see ablations in Table 3). We suspected that the issue might be due to the regularisation / KL-divergence terms in Eq. (2). These KL-divergence terms can be written as

$$D_{\text{KL}}\left(\mathcal{N}\left(\mathbf{0},\mathbf{G}\right)\|\mathcal{N}\left(\mathbf{0},\mathbf{K}\right)\right) = \text{Tr}(\mathbf{K}^{-1}\mathbf{G}) - \text{logdet}(\mathbf{K}^{-1}\mathbf{G}) + \text{const.} \tag{5}$$

This should be understood as a function, with two arguments, $\mathbf{G}$ and $\mathbf{K}$. To evaluate the objective (Eq. 2), we would set $\mathbf{G} = \mathbf{G}^{\ell}$, and $\mathbf{K} = \mathbf{K}(\mathbf{G}^{\ell})$. The KL divergence is problematic in terms of stability for two reasons. Firstly, the log-determinant term is a highly unstable operation, particularly in the backward pass which involves inverting the kernel matrix (Petersen & Pedersen, 2012). Secondly, computing $\mathbf{K}^{-1}\mathbf{G}$ for the trace requires a forward and backward substitution using the cholesky of $\mathbf{K}$, which is typically a very ill-conditioned kernel matrix.

To reduce the number of unstable operations, we replaced the log-determinant and trace terms with their second-order Taylor expansions. Since we expect the Gram representations to be close to those of the NNGP, i.e. $\mathbf{G}^{-1}\mathbf{K} \approx \mathbf{I}$, our Taylor expansions are taken around $\lambda_i = 1$, where $\lambda_i$ is the $i$th eigenvalue of $\mathbf{G}^{-1}\mathbf{K}$. In particular, the log-determinant term can be approximated as,

$$-\text{logdet}(\mathbf{K}^{-1}\mathbf{G}) = \text{logdet}(\mathbf{G}^{-1}\mathbf{K}) \tag{6a}$$

$$= \sum_i \log \lambda_i \tag{6b}$$

$$\approx \sum_i (\lambda_i - 1) - \tfrac{1}{2}(\lambda_i - 1)^2 \tag{6c}$$

$$= \text{Tr}(\mathbf{G}^{-1}\mathbf{K} - \mathbf{I}) - \tfrac{1}{2}\text{Tr}\left[(\mathbf{G}^{-1}\mathbf{K} - \mathbf{I})^2\right]. \tag{6d}$$

In the "$\approx$" step we have taken the second order Taylor expansion of $\log(\lambda_i)$ around $\lambda_i = 1$, and in the final step we have used the fact that the trace of a matrix is equal to the sum of its eigenvalues. Similarly for the trace term we have,

$$\text{Tr}(\mathbf{K}^{-1}\mathbf{G}) = \sum_i \frac{1}{\lambda_i} \tag{7a}$$

$$\approx \sum_i 1 - (\lambda_i - 1) + (\lambda_i - 1)^2 \tag{7b}$$

$$= -\text{Tr}(\mathbf{G}^{-1}\mathbf{K} - \mathbf{I}) + \text{Tr}\left[(\mathbf{G}^{-1}\mathbf{K} - \mathbf{I})^2\right] + \text{const.} \tag{7c}$$

Putting these approximations together we obtain,

$$D_{\text{KL}}\left(\mathcal{N}\left(\mathbf{0},\mathbf{G}\right)\|\mathcal{N}\left(\mathbf{0},\mathbf{K}\right)\right) \approx \tfrac{1}{2}\text{Tr}\left[(\mathbf{G}^{-1}\mathbf{K} - \mathbf{I})^2\right] + \text{const} = \tfrac{1}{2}|\mathbf{G}^{-1}\mathbf{K} - \mathbf{I}|_{\mathcal{F}}^2 + \text{const}, \tag{8}$$

where $|\cdot|_{\mathcal{F}}$ is the Frobenius norm. Computing $\mathbf{G}^{-1}\mathbf{K}$ should be more stable than $\mathbf{K}^{-1}\mathbf{G}$ since the inverse is only backpropagated to the learnt cholesky of $\mathbf{G}$, rather than through $\mathbf{K}$ to earlier parts of the model, avoiding further compounding of round-off errors.

# 4 Experiments

## 4.1 Image classification experiments

We evaluated our method on the CIFAR-10 dataset (Krizhevsky & Hinton, 2009), containing $60,000$ RGB images ($50,000$ train, $10,000$ test) of size $32 \times 32$ divided into 10 classes. We use the same

| Method | Test Accuracy (%) | Test Log-Likelihood |
|---|---|---|
| Conv. Deep Kernel Machine (This Paper) | $94.52 \pm 0.0693$ | $-0.3611 \pm 0.0073$ |
| Conv. Deep Kernel Machine (Milsom et al., 2024) | $92.69 \pm 0.0600$ | $-0.6502 \pm 0.0125$ |
| Tuned Myrtle10 Kernel DA CG (Adlam et al., 2023) | $91.2$ | - |
| NNGP-LAP-flip (Li et al., 2019) | $88.92$ | - |
| Neural Network (Adam) | $94.55 \pm 0.0361$ | $-1.3003 \pm 0.0226$ |
| Neural Network (SGD + Weight Decay) | $95.36 \pm 0.0523$ | $-0.2112 \pm 0.0037$ |

Table 1: Test metrics on CIFAR-10 using a DKM and a neural network with the same architecture. We report means and 1 standard error over 4 random seeds.

architecture as in Milsom et al. (2024) for ease of comparison, i.e. a ResNet20-inspired architecture with an extra size-2 stride in the first block, so that the output feature map sizes of the 3 blocks are $\{16, 8, 4\}$ respectively. Wherever the ResNet architecture contains a convolution layer, we use a convolutional deep kernel machine layer as described in the loop in Algorithm 1. That is, we apply a base kernel (in our case, the normalised Gaussian kernel described in Shankar et al. (2020), a more efficient / numerically stable alternative to arccos kernels), perform the (kernel) convolution, and then predict the train/test Gram matrix blocks conditioned on the kernel and the inducing Gram matrix block. In place of batch norm we use the batch kernel normalisation approach suggested by Milsom et al. (2024). Skip connections compute convex combinations of the kernel before and after pairs of layers. At the final layer, we average over the remaining spatial locations, forming an additive GP kernel akin to convolutional GPs (van der Wilk et al., 2017) that is used to make the final predictions. A categorical likelihood function is used in the top-layer GP. Since the number of inducing points can vary between layers, we use $\{512, 1024, 2048\}$ inducing points in the three blocks of convolutions, respectively, giving more expressive power to the later layers (similar to how ResNets are wider in later layers). For the stochastic kernel regularisation, we used $\gamma = P_{\mathrm{i}}^{\ell}/4$ and a jitter size of $\lambda = 0.1$, and for the objective we used a regularisation strength of $\nu = 0.001$. We train all parameters by optimising the sparse DKM objective function (Equation 27 with Taylor approximated terms from Section 3.2) using Adam (Kingma & Ba, 2017), with $\beta_1 = 0.8$, $\beta_2 = 0.9$ and with an initial learning rate of 0.01 which is divided by 10 at epochs 800 and 1100, for a total of 1200 epochs. The model is implemented[1] in PyTorch (Paszke et al., 2019).

We also train a neural network with the same architecture for comparison, using a modified version of the popular "pytorch-cifar" GitHub repository[2]. In the interest of fair comparison, we use network widths of $\{512, 1024, 2048\}$ in the three blocks so that the model has a comparable number of parameters to the convolutional deep kernel machine. The neural network was trained for 1200 epochs, and we report results using two different optimisers. One model used Adam with an initial learning rate of 0.001 and the same learning rate scheduling as the convolutional deep kernel machine, and the other used SGD with a momentum term of 0.9, a weight decay strength of 0.0005, an initial learning rate of 0.1 and a cosine annealing learning rate scheduler. We ran all experiments with 4 random seeds, and report the results in Table 1 with 1 standard error of the mean, assuming normally distributed errors for statistical tests. On an NVIDIA A100 with TF32 matmuls and convolutions enabled, the Adam-trained neural network takes ~45s per epoch, whilst our model takes ~260s per epoch. We estimate (very roughly) a total time, including the ablations and CIFAR-100 experiments detailed later, of around 2000 GPU hours for all experiments in this paper, and around 2-3 times that number when including preliminary and failed experiments during the entire project.

The deep kernel machine matches the Adam-trained neural network with a mean test accuracy of $94.52\%$ compared to $94.55\%$ for the neural network (two-tailed t-test with unequal variances gives a p-value of 0.7634, suggesting no significant difference). Furthermore, the deep kernel machine provides better uncertainty quantification as measured by (mean) log-likelihood on the test data (higher is better), with an average of $-0.3611$ compared to the Adam-trained neural network's $-1.3003$. Our model also far surpasses the convolutional deep kernel machine presented in Milsom et al. (2024). However, all these models still lag behind the SGD-trained network, which achieves a higher test accuracy of $95.36\%$ (p-value of 0.0001 when compared to our model) and higher test

---

[1]Code available at `https://github.com/edwardmilsom/skr_cdkm`

[2]`https://github.com/kuangliu/pytorch-cifar`

| Method | Test Accuracy (%) | Test Log-Likelihood |
|---|---|---|
| Conv. Deep Kernel Machine (This Paper) | $75.31 \pm 0.0814$ | $-1.4652 \pm 0.0183$ |
| Conv. Deep Kernel Machine (Milsom et al., 2024) | $72.05 \pm 0.2300$ | $-2.0553 \pm 0.0207$ |
| Neural Network (AdamW) | $74.13 \pm 0.0442$ | $-1.9183 \pm 0.0070$ |
| Neural Network (SGD + Weight Decay) | $79.42 \pm 0.0380$ | $-0.8890 \pm 0.0021$ |

Table 2: Test metrics on CIFAR-100 using a DKM and a neural network with the same architecture. We report means and 1 standard error over 4 random seeds.

| Ablation | Test Accuracy | Test Log-Likelihood | Failures |
|---|---|---|---|
| No ablation/ Our full method | $94.52 \pm 0.0693$ | $-0.3611 \pm 0.0073$ | 0/4 |
| No Taylor approximation to objective | $94.46 \pm 0.0406$ | $-0.3951 \pm 0.0081$ | 1/4 |
| No SKR | $93.71 \pm 0.0150$ | $-0.4512 \pm 0.0168$ | 2/4 |
| No Taylor + No SKR | 93.25 (1 run) | $-0.5113$ (1 run) | 3/4 |
| No SKR but keep $\lambda = 0.1$ jitter | 93.46 (1 run) | $-0.4762$ (1 run) | 3/4 |
| $\nu_\ell = 0$ (Eq. 2) | Fail | Fail | 4/4 |
| 200 epochs | $93.45 \pm 0.0225$ | $-0.2607 \pm 0.0016$ | 0/4 |

Table 3: Test metrics on CIFAR-10 with different ablations applied to our headline model (Table 1). We report means and 1 standard error over the random seeds that ran to completion. Failures indicates how many of the 4 random seed runs for each setting resulted in a numerical error.

log-likelihood of $-0.2112$ (p-value 0.00002 when compared to our model). SGD is well known to train neural networks with better generalisation properties, and in particular for ResNets (Zhou et al., 2021; Keskar & Socher, 2017; Gupta et al., 2021), so this is perhaps not too surprising. We briefly experimented with using SGD to optimise the deep kernel machine but found it generally less stable than Adam. We hypothesise this is because the deep kernel machine has many different "types" of parameters to optimise, as seen in Algorithm 1, which may benefit from different optmisation strategies, whilst the neural network only has weights and a few batchnorm parameters to optimise.

We further evaluated our method on the CIFAR-100 dataset (Krizhevsky & Hinton, 2009), with results being presented in Table 2. As in CIFAR-10, we found significant improvements over previous deep kernel machine work (Milsom et al., 2024), and we found our method is competitive with a ResNet trained with Adam, but still lags behind a ResNet trained with SGD, which is known to perform excellently on these tasks (Zhou et al., 2021; Keskar & Socher, 2017; Gupta et al., 2021). Note that we additionally had to use weight decay and a cosine annealing learning rate schedule with the Adam-trained ResNet to obtain acceptable performance on CIFAR-100.

To further investigate the effects of our changes, we ran a series of ablation experiments that are presented in Table 3. We report test accuracies and test log-likelihoods, but also the number of times each ablation failed out of the 4 random seeds as a proxy for numerical stability. Our experiments verified that stochastic kernel regularisation (SKR) did yield a statistically significant improvement in test accuracy (p-value 0.0009). To verify that the improvement was in fact coming from the random sampling of matrices and not an implicit regularising effect of the large amount of jitter, we tested the model with SKR disabled but still applying the jitter $\lambda$. We found that performance was still far worse than with SKR enabled; only 1 seed ran to completion without a numerical error for this setting, so we cannot compute the standard deviation necessary for the t-test, but based on the other experiments it is very unlikely the variance would be high enough for this not to be statistically significantly lower than our headline number. Furthermore, our Taylor approximation in the objective function did not harm performance. In fact, on log-likelihoods we obtain a p-value of 0.02953, suggesting a statistically significant improvement when using our Taylor approximated objective, but we believe this would require further investigation to verify. We also tested training with only 200 epochs, scheduling down the learning rate at epochs 160 and 180, and found that training for a 1200 epochs did indeed give a substantial boost to test accuracy. We found no single trick was enough to ensure stability over all our training runs, but rather a combination of our proposed modifications was necessary. We provide some brief analysis of the stability of the learned Gram matrices in the next section.

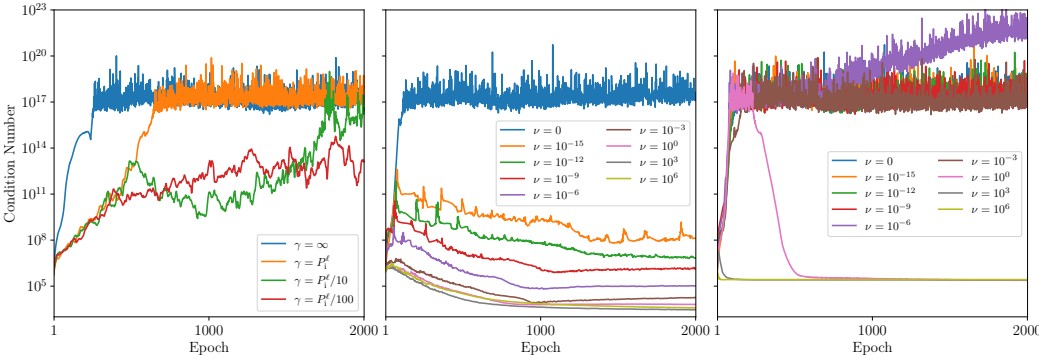

Figure 1: Effects of different regularisation methods on Gram matrix condition number, in the toy binary classification problem trained for 2000 epochs. The left plot shows the condition numbers when different amounts of stochastic kernel regularisation ($\gamma$) are applied. The middle and right plots show the condition numbers when the coefficient $\nu$ of the KL regularisation terms are varied, with and without a Taylor approximation, respectively.

## 4.2    Effects of regularisation on Gram matrix condition number

To further investigate the numerical stability of our model, we ran a 1 layer deep kernel machine with the squared exponential kernel and 100 inducing points on a toy binary classification problem. We show the condition number of the learned Gram matrix at each epoch for various values of the SKR parameter $\gamma$ (left, Fig. 1), the DKM objective regularisation strength $\nu$ when using the Taylor KL divergence terms (middle, Fig. 1), and $\nu$ when using the true KL divergence (right, Fig. 1). For the plot varying $\gamma$, we used $\nu = 0$, so that the effect of the Taylor approximation to the KL terms is irrelevant. Note that $\gamma = \infty$ refers to no SKR. We ran these experiments in double precision to avoid numerical errors, and used the Adam optimiser with learning rate fixed at 0.01. These experiments took about 1 CPU hour in total.

We observe that more variance (smaller $\gamma$) in stochastic kernel regularisation slows down the rate at which the condition number of $\mathbf{G}_{ii}$ worsens. This makes sense, as the noise we add to the Gram matrix makes it more difficult to learn the "optimal" Gram matrix for the data, which would likely be highly overfitted, leading to extreme eigenvalues. However, running the experiment for long enough eventually leads to the same condition number for all settings of $\gamma$ (see Fig 2 in Appendix B). We may expect this behaviour since the expected value of the SKR samples matches the true Gram matrix, but it's not clear how effectively the optimiser could achieve this outside of simple toy examples.

It is clear that, when using our proposed Taylor approximation to the KL divergence terms in the objective, even tiny values for the strength $\nu$ of these terms result in learned Gram matrices with condition numbers orders of magnitude better than without, and this effect grows proportionally with $\nu$. We also see an improvement in condition number when not using the Taylor approximation to the KL divergence, but only for large $\nu$. Setting $\nu$ too large tends to harm generalisation (Milsom et al., 2024), so it is beneficial to use the Taylor approximation which reduces condition numbers even for small $\nu$. We also observed that the minimum condition number achieved across all settings of $\nu$ was a few orders of magnitude lower when using the Taylor approximation vs. when using the true KL divergence. Furthermore, the behaviour in the plot using the true KL divergence is rather erratic. For example, the $\nu = 10^0$ curve (right, Fig. 1) initially rises to very poor condition numbers, but after a few hundred epochs rapidly drops to a much smaller condition number. This leads us to believe that the difference between these two schemes can be explained by optimisation.

Our Taylor approximated terms penalise the Frobenius norm $|\mathbf{GK} - \mathbf{I}|_{\mathcal{F}}^2$, a much simpler operation than the true KL divergence terms which penalise $\text{Tr}\left(\mathbf{G}^{-1}\mathbf{K}\right) - \text{logdet}\left(\mathbf{G}^{-1}\mathbf{K}\right)$. This complex penalty may result in a difficult optimisation landscape in practice.

# 5    Related Work

There is a substantial body of literature attempting to push the performance of kernel methods to new heights. These methods can broadly be split into "kernel learning" and "fixed kernel" methods.

Deep kernel machines, already extensively discussed in this paper, fall into the former category, as does the area of "deep kernel learning" (e.g. Wilson et al., 2016; Achituve et al., 2023; Ober et al., 2021, to name a few). In deep kernel learning, a neural network is used to produce rich features which are then passed as inputs into a traditional "shallow" kernel machine, aiming to give the best of both deep learning and kernel methods. Another "kernel learning" method is the "convolutional kernel machine" (Mairal et al., 2014; Mairal, 2016), which draw theoretical connections between kernel methods and neural networks, though the resulting model is fundamentally a neural-network-like architecture based on features, which distinguishes it from deep kernel machines. Song et al. (2017) also utilised deep neural networks to generate task-specific representations in a more complex model involving an ensemble of RKHS subspace projections. The main difference between deep kernel machines and these other methods is that deep kernel machines do not involve neural networks at any stage; the representations are learned directly as Gram matrices, not features.

By contrast, "fixed kernel" methods do not perform any representation learning during training, instead fixing their feature space before data is seen via the choice of kernel function. Though this could cover practically the entire field of kernel methods, the best performing methods on image tasks typically utilise kernels derived from the infinite-width neural network literature (Lee et al., 2017; Jacot et al., 2018; Lee et al., 2020), sometimes called "neural kernels" (Shankar et al., 2020). In particular, Adlam et al. (2023) pushed the state of the art for CIFAR-10 test accuracy with "fixed kernels" to 91.2%, using Myrtle kernels (Shankar et al., 2020), a type of neural kernel, by massively scaling up their method with distributed preconditioned conjugate gradient methods. Apart from the obvious lack of representation learning in this work, another key difference from our work is that they focus on computing large full-rank kernel matrices and finding approximate solutions using iterative solvers, whilst we use sparse inducing point approximations resulting in smaller kernel matrices, which we then solve exactly.

Deep kernel machines can be viewed as an infinite-width limit of deep kernel processes (Yang et al., 2023) or deep Gaussian processes (Damianou & Lawrence, 2013) with a modified likelihood function, which results in the Gram matrices having point distributions. This can lead to overfitting. In deep Gaussian processes and deep kernel processes, the representations (Gram matrices in the case of deep kernel processes) have continuous distributions with broad support, thereby offering a regularising effect. Our stochastic kernel regularisation scheme can be seen as analogous to sampling the inducing Gram matrix $\mathbf{G}_{ii}^{\ell}$ in a deep kernel process. Unlike a deep kernel process, the other blocks $\mathbf{G}_{ti}^{\ell}$ and $\mathbf{G}_{tt}^{\ell}$ in our model remain deterministic, simplifying the model implementation. Other approaches to regularising kernel methods include "kernel dropout" proposed by Song et al. (2017), though in their context dropout refers to randomly removing latent representations from their ensemble during training. This is therefore very different to our setting. In the neural kernel literature, Lee et al. (2020) identified a correspondence between diagonal regularisation of kernels (jitter) and early stopping in neural networks, and found this usually improved generalisation. In this paper, we focused on regularising the learned intermediate representations / Gram matrices, rather than the final kernel, and found that diagonal regularisation had little effect on generalisation when applied to these matrices.

Previous work has attempted to improve numerical stability in kernel methods, though using different approaches. For example, Maddox et al. (2022) developed strategies to ensure numerical stability when using conjugate gradient solvers for GPs with low-precision arithmetic, but we do not use numerical solvers in this paper. van der Wilk et al. (2020) circumvent the issue of computing inverses entirely using an approach based on a reparametrisation of the variational parameters, but applying such an approach to the deep kernel machine domain would be a substantial undertaking, which we leave to future work.

# 6    Limitations

Though we have considerably advanced the state-of-the-art for kernel methods, from 92.7% (Milsom et al., 2024) to 94.5% in CIFAR-10, there still remains a gap to the best performing neural networks, both in terms of accuracy, and in terms of runtime. Nonetheless, given that we have shown that

representation learning in kernel methods has dramatically improved performance in kernel methods, from 91.2% (Adlam et al., 2023) to 94.5%, it is becoming increasingly likely that representation learning really is the key reason that NNGPs underperform DNNs. We leave further narrowing or even closing the remaining gap to DNNs for future work.

Constraints on computational resources meant that we could only run a limited number of experiments, so we focused on providing concrete insights on a single dataset with a series of ablations, rather than performance metrics for multiple datasets with no further analysis. Nevertheless, we provide all the code necessary to run these experiments on other datasets.

## 7 Conclusion

In this paper we have increased the kernel SOTA for CIFAR-10 to $94.5\%$ test accuracy using deep kernel machines, considerably higher than the previous record of $92.7\%$ (Milsom et al., 2024), and significantly higher than NNGP-based approaches, such as the $91.2\%$ achieved by Adlam et al. (2023). We achieved this by developing a novel regularisation method, stochastic kernel regularisation, and by exploiting modern GPU hardware with lower-precision arithmetic, which required us to improve the numerical stability of the algorithm via a multi-faceted approach. We have highlighted the important role that representation learning plays in deep learning, which is unfortunately absent from NNGP-based theory. We hope this work will encourage more research into theoretical models with representation learning.

## 8 Acknowledgements

Edward Milsom and Ben Anson are funded by the Engineering and Physical Sciences Research Council via the COMPASS Centre for Doctoral Training at the University of Bristol. This work was carried out using the computational facilities of the Advanced Computing Research Centre, University of Bristol - `http://www.bris.ac.uk/acrc/`. We would like to thank Dr. Stewart for GPU compute resources.

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

## A   Introduction to (Convolutional) Deep Kernel Machines

An introduction to both fully-connected and convolutional deep kernel machines can be found in (Milsom et al., 2024), but for completeness we provide an overview here. In particular, we show how sampling a large number of features at each layer of a deep Gaussian process gives rise to a deep kernel method.

### A.1   Fully-connected deep kernel machines

A DKM can been seen as a wide deep Gaussian process (DGP), optimised using a tempered ELBO objective. To see why, we first define a DGP where subsequent layers of features are conditionally multivariate Gaussian. Assume we have $P$ input data points $\mathbf{X} \in \mathbb{R}^{P \times N_0}$, and corresponding label categories $\mathbf{y} \in \{1, \ldots, C\}^P$, where $C$ is the number of categories. Then we can place the following DGP prior on the data,

$$\mathbf{F}^0 = \mathbf{X}, \tag{9a}$$

$$P(\mathbf{F}^\ell \mid \mathbf{F}^{\ell-1}) = \prod_{\lambda=1}^{N_\ell} \mathcal{N}(\mathbf{f}_\lambda^\ell; \mathbf{0}, \mathbf{K}_{\text{features}}(\mathbf{F}^{\ell-1})), \tag{9b}$$

$$P(\mathbf{y} \mid \mathbf{F}^{L+1}) = \prod_{i=1}^{P} \text{Categorical}(y_i; \text{softmax}((\mathbf{F}^{L+1})_{i,:})). \tag{9c}$$

Here $\mathbf{F}^\ell \in \mathbb{R}^{P \times N_\ell}$ denotes the $N_\ell$ features at layer $\ell$, and $\mathbf{f}_\lambda^\ell$ is the $\lambda$th feature at layer $\ell$. $\mathbf{K}_{\text{features}}(\cdot)$ is a kernel function that takes in features (as kernel functions usually do), written with the subscript "features" to different it later from kernel functions that take Gram matrices as input. Notice that the

final layer features $\mathbf{F}^{L+1}$ are logits for a categorical distribution over labels, though the likelihood distribution can be easily changed for the regression setting.

To derive an ELBO, we perform variational inference by defining the following approximate posterior over the features at intermediate and final layers,

$$\mathrm{P}(\mathbf{F}^{\ell} \mid \mathbf{X}, \mathbf{Y}) \approx \mathrm{Q}(\mathbf{F}^{\ell}) = \prod_{\lambda=1}^{N_{\ell}} \mathrm{Q}(\mathbf{f}_{\lambda}^{\ell}) = \prod_{\lambda=1}^{N_{\ell}} \mathcal{N}(\mathbf{f}_{\lambda}^{\ell}; \mathbf{0}, \mathbf{G}^{\ell}), \tag{10a}$$

$$\mathrm{P}(\mathbf{F}^{L+1} \mid \mathbf{X}, \mathbf{Y}) \approx \mathrm{Q}(\mathbf{F}^{L+1}) = \prod_{\lambda=1}^{N_{\ell}} \mathrm{Q}(\mathbf{f}_{\lambda}^{L+1}) = \prod_{\lambda=1}^{N_{L+1}} \mathcal{N}(\mathbf{f}_{\lambda}^{L+1}; \boldsymbol{\mu}_{\lambda}, \boldsymbol{\Sigma}). \tag{10b}$$

We learn the mean and covariance at the final layer, but only the covariances $\mathbf{G}^{\ell} \in \mathbb{R}^{P \times P}$ at the intermediate layers. This choice is justified by the fact that after taking the limit $N_{\ell} \to \infty$, this approximate posterior family (Eq. 10) contains the true posterior (see Yang et al. (2023) for more details). The ELBO of the DGP with respect to the variational parameters is,

$$\mathcal{L}_{\text{ELBO}}(\mathbf{G}_1, \ldots, \mathbf{G}_L, \boldsymbol{\mu}_1, \ldots, \boldsymbol{\mu}_{N_{L+1}}, \boldsymbol{\Sigma}) = \tag{11}$$

$$\mathbb{E}_{\mathrm{Q}(\mathbf{F}^{L+1})} \big[ \log \mathrm{P}(\mathbf{y} \mid \mathbf{F}^{L+1}) \big] - \sum_{\lambda=1}^{N_{L+1}} \mathrm{D}_{\text{KL}}(\mathrm{Q}(\mathbf{f}_{\lambda}^{L+1}) \,\|\, \mathrm{P}(\mathbf{f}_{\lambda}^{L+1} \mid \mathbf{F}^{L}))$$

$$- \sum_{\ell=1}^{L} \beta N_{\ell} \mathrm{D}_{\text{KL}}(\mathrm{Q}(\mathbf{f}^{\ell}) \,\|\, \mathrm{P}(\mathbf{f}^{\ell} \mid \mathbf{F}^{\ell-1})). \tag{12}$$

Here $\beta$ is parameter that tempers the prior. Before proceeding, we make the following assumption that the kernel can be calculated using only the sample feature covariance $\hat{\mathbf{G}}^{\ell}$:

$$\mathbf{K}_{\text{features}}(\mathbf{F}^{\ell}) = \mathbf{K}(\hat{\mathbf{G}}^{\ell}), \tag{13a}$$

$$\text{where } \hat{\mathbf{G}}^{\ell} = \tfrac{1}{N_{\ell}} \mathbf{F}^{\ell}(\mathbf{F}^{\ell})^{T}. \tag{13b}$$

Assumption 13 is actually not very restrictive — it is satisfied by common kernels (such as RBF, Matern), and indeed any isotropic kernel. It is also satisfied in the limit $N_{\ell} \to \infty$ when $\mathbf{K}_{\text{feature}}(\mathbf{X}) = \text{ReLU}(\mathbf{X})\text{ReLU}(\mathbf{X})^{T}$ by the arccosine kernel (Cho & Saul, 2009).

We are now ready to recover a DKM. We set $N_{\ell} = N\nu_{\ell}$ for each intermediate layer $\ell = 1, \ldots, L$, and temper with $\beta = N^{-1}$. In the limit $N \to \infty$, Yang et al. (2023) showed that the limiting ELBO is,

$$\mathcal{L}_{\text{ELBO}} \to \mathcal{L} := \mathbb{E}_{\mathrm{Q}(\mathbf{F}^{L+1})} \big[ \log \mathrm{P}(\mathbf{y} \mid \mathbf{F}^{L+1}) \big] - \sum_{\lambda=1}^{N_{L+1}} \mathrm{D}_{\text{KL}}(\mathcal{N}(\boldsymbol{\mu}_{\lambda}, \boldsymbol{\Sigma}) \,\|\, \mathcal{N}(\mathbf{0}, \mathbf{K}(\mathbf{G}^{L}))$$

$$- \sum_{\ell=1}^{L} \nu_{\ell} \mathrm{D}_{\text{KL}}(\mathcal{N}(\mathbf{0}, \mathbf{G}^{\ell}) \,\|\, \mathcal{N}(\mathbf{0}, \mathbf{K}(\mathbf{G}^{\ell-1}))), \tag{14}$$

where $\mathbf{K}(\cdot) : \mathbb{R}^{P \times P} \to \mathbb{R}^{P \times P}$ a kernel function satisfying Assumption 13. We can evaluate the expected log-likelihood in the DKM objective (Eq. (14)) using the reparameterisation trick, and the KL divergence terms can be calculated in closed form. By optimizing $\mathcal{L}$ w.r.t. the variational parameters we 'train' the DKM. Optimisation is not possible in closed form in general (it is possible in the linear kernel case for regression problems, see Yang et al. (2023)), but we can train the parameters using gradient descent. The number of parameters is $\mathcal{O}(P^2)$ and the time complexity for evaluating the objective is $\mathcal{O}(P^3)$, therefore we can only optimise Eq. (14) directly for small datasets. We later discuss an inducing point method that enables linear scaling in the number of datapoints, but first we introduce DKMs with convolutions.

## A.2 Convolutional deep kernel machines

Above we outlined how a fully-connected DKM is a DGP with wide intermediate layers trained by tempering the ELBO. We establish a DKM for the convolutional setting by introducing a convolution

into the DGP in Eq. (9),

$$\mathbf{F}^0 = \mathbf{X}, \tag{15a}$$

$$P(\mathbf{H}^\ell \mid \mathbf{F}^\ell) = \prod_{\lambda=1}^{M_\ell} \mathcal{N}(\mathbf{h}_\lambda^\ell; \mathbf{0}, \mathbf{K}_{\text{features}}(\mathbf{F}^\ell)), \tag{15b}$$

$$W_{d\mu\lambda}^\ell \sim_{\text{iid}} \mathcal{N}(0, (|\mathcal{D}|\, M_{\ell-1})^{-1}), \tag{15c}$$

$$F_{ir,\lambda}^\ell = \sum_{d\in\mathcal{D}} \sum_{\mu=1}^{M_{\ell-1}} H_{i(r+d),\mu}^{\ell-1} W_{d\mu\lambda}^\ell, \tag{15d}$$

for $\ell = 1, \ldots, L$, as well as a spatial pooling layer before final classification,

$$\mathbf{F}_{\text{flat}}^L = \text{SpatialPool}(\mathbf{F}^L) \tag{15e}$$

$$P(\mathbf{F}^{L+1} \mid \mathbf{F}^L) = \prod_{\lambda=1}^{N_{L+1}} \mathcal{N}(\mathbf{f}_\lambda^{L+1}; \mathbf{0}, \mathbf{K}_{\text{features}}(\mathbf{F}_{\text{flat}}^L)), \tag{15f}$$

$$P(\mathbf{y} \mid \mathbf{F}^{L+1}) = \prod_{i=1}^{P} \text{Categorical}(y_i; \text{softmax}((\mathbf{F}^{L+1})_{i,:})). \tag{15g}$$

Here, we consider datapoints to be spatial locations (a.k.a. patches) across a range of images, and we index these with $i$ (image) and $r$ (location). By concatenating all patches (i.e. every patch in every image) together, we can represent an entire dataset of images with a single feature matrix $\mathbf{F}^\ell \in \mathbb{R}^{P|\mathcal{S}| \times N_\ell}$, where $\mathcal{S}$ is the set of patches in an image and $P$ is the number of images. For us, $\mathbf{y}$ is a set of image-level labels rather than patch-level labels, so we also include a spatial pooling layer. The pooled features $\mathbf{F}_{\text{flat}}^L$ have size $P \times N_\ell$. The convolution (Eq. 15d) uses convolutional weights $\mathbf{W}^\ell \in \mathbb{R}^{|\mathcal{D}| \times M_{\ell-1} \times N_\ell}$, where $\mathcal{D}$ is the set of spatial locations in the filter. In our context, we only consider 2-dimensional images, therefore $\mathcal{D}$ will contain $|\mathcal{D}|$ 2-dimensional locations of patches.

We proceed by deriving a convolutional kernel. The conditional covariance of the features $\mathbf{F}^\ell$ has closed form,

$$\mathbb{E}[F_{ir,\mu}^\ell F_{js,\mu'}^\ell \mid \mathbf{H}^{\ell-1}] = \mathbb{E}\left[ \sum_{d\mu} H_{i(r+d)\mu}^{\ell-1} W_{d\mu,\lambda}^\ell \sum_{d'\mu'} H_{j(s+d')\mu'}^{\ell-1} W_{d'\mu',\lambda}^\ell \right] \tag{16}$$

$$= \sum_{dd'\mu\mu'} H_{i(r+d)\mu}^{\ell-1} H_{j(s+d')\mu'}^{\ell-1} \mathbb{E}\left[ W_{d\mu,\lambda}^\ell W_{d'\mu',\lambda}^\ell \right] \tag{17}$$

$$= \frac{1}{DM_\ell} \sum_{d\in\mathcal{D}} \sum_{\mu=1}^{M_\ell} H_{i,(r+d)\mu}^{\ell-1} H_{j,(s+d)\mu'}^{\ell-1} \tag{18}$$

$$= \frac{1}{D} \sum_{d\in\mathcal{D}} \hat{\Omega}_{i(r+d),j(s+d)}^{\ell-1}, \tag{19}$$

where $\hat{\boldsymbol{\Omega}}^\ell$ is the sample covariance of $\mathbf{H}^\ell$. Under the layer-wise, infinite-width limit $M_\ell \to \infty$, the sample covariance becomes the true covariance, $\hat{\boldsymbol{\Omega}}^\ell \to \mathbf{K}_{\text{features}}(\mathbf{F}^\ell)$. This means that we can compute the covariance of $\mathbf{F}^\ell$ conditioned only on the previous layer,

$$\mathbb{E}[F_{ir,\mu}^\ell F_{js,\mu'}^\ell \mid \mathbf{F}^{\ell-1}] = \frac{1}{D} \sum_{d\in\mathcal{D}} (\mathbf{K}_{\text{features}}(\mathbf{F}^{\ell-1}))_{i(r+d),j(s+d)}. \tag{20}$$

We can view Eq. 20 as a kernel convolution operation, and we introduce the following notation for it: $(\boldsymbol{\Gamma}(\mathbf{K}))_{ir,js} = \frac{1}{D} \sum_{d\in\mathcal{D}} K_{i(r+d),j(s+d)}$.

Equipped with a convolutional kernel $\boldsymbol{\Gamma}$, we can recover a convolutional deep kernel machine by taking the limit $N_\ell \to \infty$. We again use the approximate posterior defined in Eq. 10 and temper the

prior. This gives us the convolutional DKM objective,

$$\mathcal{L} := \mathbb{E}_{Q(\mathbf{F}^{L+1})}\big[\log P(\mathbf{y} \mid \mathbf{F}^{L+1})\big] - \sum_{\lambda=1}^{N_{L+1}} D_{KL}(\mathcal{N}(\boldsymbol{\mu}_\lambda, \boldsymbol{\Sigma}) \parallel \mathcal{N}(\mathbf{0}, \mathbf{K}(\text{SpatialPool}(\mathbf{G}^L))))$$

$$- \sum_{\ell=1}^{L} \nu_\ell D_{KL}(\mathcal{N}(\mathbf{0}, \mathbf{G}^\ell) \parallel \mathcal{N}(\mathbf{0}, \boldsymbol{\Gamma}(\mathbf{K}(\mathbf{G}^{\ell-1})))), \qquad (21)$$

where again we used Assumption 13. The spatial pool operation used in this paper is mean pooling (see Algorithm 1). This convolutional DKM objective implies a fixed convolutional kernel, but allows flexibility at intermediate layers via the variational parameters $\mathbf{G}^\ell$.

### A.3 Inducing point approximations

Due to $\mathcal{O}(P^2)$ parameters, and $\mathcal{O}(P^3)$ computational cost, it is not feasible to optimise the DKM objectives (Eq. 14 and Eq. 21) directly. To resolve this scaling problem, we appeal to inducing point methods. The idea is to approximate the full train/test dataset with a smaller set of datapoints. We demonstrate an inducing point convolutional DKM here. See Appendix M in Yang et al. (2023) for the fully-connected case.

We define the train/test datapoint features $\mathbf{F}_t^\ell \in \mathbb{R}^{P_t \times N_\ell}$, and the inducing datapoints $\mathbf{F}_i^\ell \in \mathbb{R}^{P_i^\ell \times N_\ell}$; $P_t$ is the number of train/test datapoints and $P_i^\ell$ is the number of inducing points at layer $\ell$. We take the inducing points and train/test points to be jointly distributed according to the deep Gaussian process in Eq. 15. In other words, the concatenation of features

$$\mathbf{F}^\ell = \begin{pmatrix} \mathbf{F}_i^\ell \\ \mathbf{F}_t^\ell \end{pmatrix} \in \mathbb{R}^{(P_i^\ell + P_t) \times N_\ell} \qquad (22)$$

satisfies the prior in Eq. 15, with the exception that the convolution for the inducing points is slightly modified so that,

$$(F_i^\ell)_{i,\lambda} = \sum_{d \in \mathcal{D}} \sum_{\mu=1}^{M_\ell} W_{d\mu,\lambda}^\ell \sum_{i'} C_{di,i'}^\ell (H_i^{\ell-1})_{i',\mu}. \qquad (23)$$

Milsom et al. (2024) motivated the extra 'mixup' parameter $\mathbf{C}^\ell \in \mathbb{R}^{|\mathcal{D}|P_i^\ell \times P_i^{\ell-1}}$ as allowing informative correlations between the inducing points and the train/test points to be learned. We can view the matrix multiplication $\mathbf{C}^\ell \mathbf{H}_i^\ell$ in Eq. 23 as having the effect of taking non-spatial inducing points $\mathbf{H}_i^\ell$ and mapping to them to inducing patches. This allows them to correlate meaningfully with the train/test patches. The covariances among inducing and train/test features can then be calculated,

$$\mathbb{E}\big[(F_i^\ell)_{i,\lambda}(F_i^\ell)_{j,\lambda} \mid \mathbf{H}^{\ell-1}\big] = \frac{1}{D} \sum_{d \in \mathcal{D}} \sum_{i'} \sum_{j'} C_{di,i'}^\ell C_{dj,j'}^\ell (\hat{\Omega}_{ii}^{\ell-1})_{i'j'} := (\Gamma_{ii}^\ell(\hat{\boldsymbol{\Omega}}^{\ell-1}))_{i,j}, \quad (24a)$$

$$\mathbb{E}\big[(F_i^\ell)_{i,\lambda}(F_t^\ell)_{js,\lambda} \mid \mathbf{H}^{\ell-1}\big] = \frac{1}{D} \sum_{d \in \mathcal{D}} \sum_{i'} C_{di,i'}^\ell (\hat{\Omega}_{it}^{\ell-1})_{i',j(s+d)} := (\Gamma_{it}^\ell(\hat{\boldsymbol{\Omega}}^{\ell-1}))_{i,js} \qquad (24b)$$

$$\mathbb{E}\big[(F_t^\ell)_{is,\lambda}(F_t^\ell)_{jv,\lambda} \mid \mathbf{H}^{\ell-1}\big] = \frac{1}{D} \sum_{d \in \mathcal{D}} (\hat{\Omega}_{tt}^{\ell-1})_{i(s+d),j(v+d)} := (\Gamma_{ti}^\ell(\hat{\boldsymbol{\Omega}}^{\ell-1}))_{is,jv}. \qquad (24c)$$

Here, $\hat{\boldsymbol{\Omega}}^\ell = \frac{1}{M_\ell} \mathbf{H}^\ell(\mathbf{H}^\ell)^T$ is the sample covariance for combined inducing and train/test samples $\mathbf{H}^\ell = \begin{bmatrix} \mathbf{H}_i^\ell & \mathbf{H}_t^\ell \end{bmatrix}^T$. Suffices ii refer to inducing-inducing correlations, ti to train/test-inducing correlations, and tt refer to train/test-train/test correlations; though they may better be understood as referring to different blocks of a covariance matrix. Eq. 24 gives us a learnable convolutional kernel, which we call $\boldsymbol{\Gamma}^\ell(\cdot)$. When we take the convolutional widths $M_\ell$ to be large, we have,

$$P(\mathbf{F}^\ell \mid \mathbf{F}^{\ell-1}) = \prod_{\lambda=1}^{N_\ell} \mathcal{N}(\mathbf{f}_\lambda^\ell; \mathbf{0}, \boldsymbol{\Gamma}^\ell(\mathbf{K}_{\text{features}}(\mathbf{F}_{\ell-1}))). \qquad (25)$$

As in the non-inducing case, we will compute an ELBO. To do so, we place an approximate posterior on the inducing points, similar to Eq. 10,

$$Q(\mathbf{F}_i^\ell) = \prod_{\lambda=1}^{N_\ell} Q(\mathbf{f}_{i;\lambda}^\ell) = \prod_{\lambda=1}^{N_\ell} \mathcal{N}(\mathbf{f}_{i;\lambda}^\ell \mathbf{0}, \mathbf{G}_{ii}^\ell), \tag{26a}$$

$$Q(\mathbf{F}_i^{L+1}) = \prod_{\lambda=1}^{N_\ell} Q(\mathbf{f}_{i;\lambda}^{L+1}) = \prod_{\lambda=1}^{N_{L+1}} \mathcal{N}(\mathbf{f}_{i;\lambda}^{L+1}; \boldsymbol{\mu}_\lambda, \boldsymbol{\Sigma}). \tag{26b}$$

Taking the layer-wise infinite-width limit $N_\ell \to \infty$ (again tempering the prior as in Eq. 14), we recover the following limit of the ELBO,

$$\mathcal{L}_{\text{inducing}} := \mathbb{E}_{Q(\mathbf{F}^{L+1})}\big[\log P(\mathbf{y} \mid \mathbf{F}_t^{L+1})\big] \tag{27a}$$

$$- \sum_{\lambda=1}^{N_{L+1}} D_{\text{KL}}(\mathcal{N}(\boldsymbol{\mu}_\lambda, \boldsymbol{\Sigma}) \parallel \mathcal{N}(\mathbf{0}, \mathbf{K}(\text{SpatialPool}(\mathbf{G}_{ii}^L))))$$

$$- \sum_{\ell=1}^{L} \nu_\ell D_{\text{KL}}(\mathcal{N}(\mathbf{0}, \mathbf{G}_{ii}^\ell) \parallel \mathcal{N}(\mathbf{0}, \boldsymbol{\Gamma}^\ell(\mathbf{K}(\mathbf{G}_{ii}^{\ell-1})))), \tag{27b}$$

However, it still remains to perform inference on the train/test points. To this end, we 'connect' the train/test points to the inducing points by assuming the approximate posterior over all features decomposes like so,

$$Q(\mathbf{F}^\ell \mid \mathbf{F}^{\ell-1}) = P(\mathbf{F}_t^\ell \mid \mathbf{F}_i^\ell, \mathbf{F}^{\ell-1})Q(\mathbf{F}_i^\ell). \tag{28}$$

Due to the Gaussian structure of the DGP prior, the first term in Eq. 28 is Gaussian and can be written down using standard conditional Gaussian expressions,

$$P(\mathbf{F}_t^\ell \mid \mathbf{F}_i^\ell, \mathbf{F}^{\ell-1}) = \prod_{\lambda=1}^{N_\ell} \mathcal{N}(\mathbf{f}_{t;\lambda}^\ell; \boldsymbol{\Gamma}_{ti}^\ell(\boldsymbol{\Gamma}_{ii}^\ell)^{-1}\mathbf{f}_{i;\lambda}^\ell, \boldsymbol{\Gamma}_{tt}^\ell - \boldsymbol{\Gamma}_{ti}^\ell(\boldsymbol{\Gamma}_{ii}^\ell)^{-1}\boldsymbol{\Gamma}_{it}^\ell), \tag{29}$$

where $\boldsymbol{\Gamma}^\ell$ is the result after applying the non-linearity kernel and then the convolutional kernel, i.e. $\boldsymbol{\Gamma}^\ell = \boldsymbol{\Gamma}(\mathbf{K}_{\text{features}}(\mathbf{F}^\ell))$. In other words, we can write $\mathbf{F}_t^\ell$ in terms of standard multivariate Gaussian noise $\boldsymbol{\Xi} \in \mathbb{R}^{P_t \times N_\ell}$,

$$\mathbf{F}_t^\ell = \boldsymbol{\Gamma}_{ti}^\ell(\boldsymbol{\Gamma}_{ii}^\ell)^{-1}\mathbf{F}_i^\ell + \boldsymbol{\Gamma}_{tt \cdot i}^{1/2}\boldsymbol{\Xi}, \tag{30a}$$

$$\text{where } \boldsymbol{\Gamma}_{tt \cdot i} = \boldsymbol{\Gamma}_{tt}^\ell - \boldsymbol{\Gamma}_{ti}^\ell(\boldsymbol{\Gamma}_{ii}^\ell)^{-1}\boldsymbol{\Gamma}_{it}^\ell. \tag{30b}$$

In the infinite-width limit $N_\ell \to \infty$, the sample feature covariance must converge to the true covariance by the law of large numbers,

$$\frac{1}{N_\ell}\mathbf{F}^\ell(\mathbf{F}^\ell)^T \to \mathbb{E}\big[\mathbf{f}^\ell(\mathbf{f}^\ell)^T\big] = \mathbf{G}^\ell = \begin{pmatrix} \mathbf{G}_{ii}^\ell & \mathbf{G}_{it}^\ell \\ \mathbf{G}_{ti}^\ell & \mathbf{G}_{tt}^\ell \end{pmatrix}. \tag{31}$$

We already know the true inducing point covariance matrices, $\mathbf{G}_{ii}^\ell$, because they are parameters in our approximate posterior. However we can write down the remaining blocks of the covariance using Eq. 30. We identify $\mathbf{G}_{ti}^\ell$ and $\mathbf{G}_{tt}^\ell$ as,

$$\mathbf{G}_{ti}^\ell = \lim_{N_\ell \to \infty} \frac{1}{N_\ell}\mathbf{F}_t^\ell(\mathbf{F}_i^\ell)^T = \boldsymbol{\Gamma}_{ti}^\ell(\boldsymbol{\Gamma}_{ii}^\ell)^{-1}\mathbf{G}_{ii}^\ell \tag{32a}$$

$$\mathbf{G}_{tt}^\ell = \lim_{N_\ell \to \infty} \frac{1}{N_\ell}\mathbf{F}_t^\ell(\mathbf{F}_t^\ell)^T = \boldsymbol{\Gamma}_{ti}^\ell(\boldsymbol{\Gamma}_{ii}^\ell)^{-1}\mathbf{G}_{ii}^\ell(\boldsymbol{\Gamma}_{ii}^\ell)^{-1}\boldsymbol{\Gamma}_{it}^\ell + \boldsymbol{\Gamma}_{tt \cdot i}. \tag{32b}$$

Equations 32 and 24, as seen in Algorithm 1, allow us to propagate train/test points through the model alongside learned inducing points.

In the above, we treat datapoints as independent. This allows us to perform minibatch training, thus greatly improving the scaling of the method over the full-rank version. Alongside the reduction in learnable parameters from the inducing point scheme, we get linear scaling with the size of the dataset.

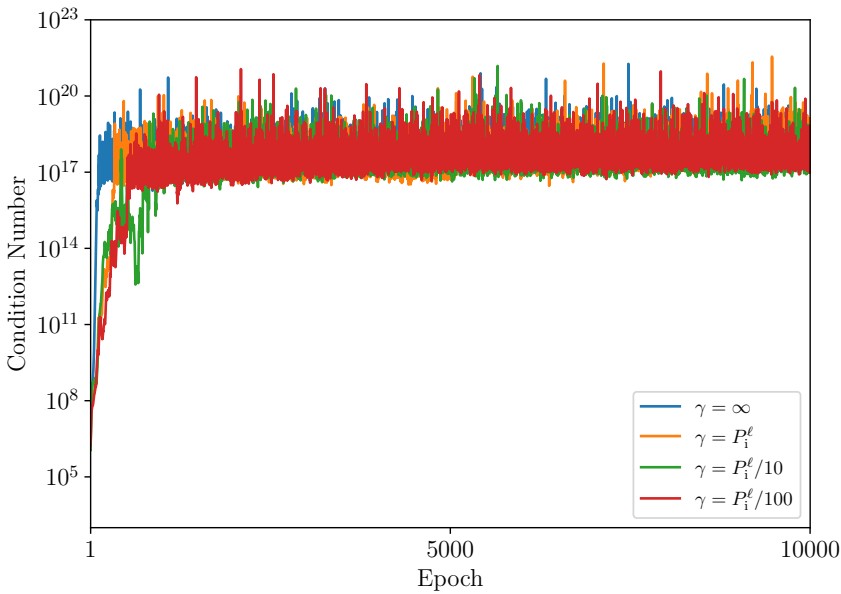

Figure 2: Effects of stochastic kernel regularisation on Gram matrix condition number strength, in the toy binary classification problem trained for 10000 epochs. See Section 4.2.

## B    Extra Figures

## C    Licenses

