# OpenReview forum: "Stochastic Kernel Regularisation Improves Generalisation in Deep Kernel Machines"
_NeurIPS.cc/2024/Conference — NeurIPS 2024 poster_

### Official Review · Reviewer_YneB · 2024-07-09

**Soundness:** 3
**Presentation:** 3
**Contribution:** 2
**Rating:** 7
**Confidence:** 2

**Summary:**

The authors present an improved convolutional deep kernel machine (CDKM) that achieves state-of-the-art performance for kernel methods on the CIFAR-10 image classification task. They introduce a novel regularization technique where the learned inducing Gram matrices are randomly sampled from a Wishart distribution during training. This helps reduce overfitting and improves generalization.

**Strengths:**

The experiments evaluating the performance of their methods are thorough and convincing in providing an improvement over previous work.

They achieve 94.52% test accuracy on CIFAR-10, which is a significant improvement over the previous best kernel method result of 92.69% and comes close to matching comparable neural network performance (94.55% for an Adam-trained neural network with the same architecture).

As shown in Table 2, their improvements in numerical stability allow for the use of TF32 arithmetic, making training about 5 times faster than previous implementations that used double-precision floating point arithmetic.

In Table 1, the test-log likelihood improvements over previous work and traditional neural networks (without weight decay) is compelling as a result. But also surprised to see models with weight decay outperforming the other methods, if I understand correctly, CDKMs naturally provide uncertainty estimates for their predictions. Why would traditional models (with weight decay) be better at this?

**Weaknesses:**

Disclaimer: I am not familiar with the space of CDKMs. It's unclear to me what the main scientific insight is provided in this work. The results appear to be state-of-the-art in this space. But aside from performance, it's unclear to me that this isn't just a variation of traditional neural networks with some aspects of Kernel Machines added.

But this might be an issue with the general space of CDKMs and not this work specifically. Since the CDKM method is using some form of learning with backpropagation, what is the main point of showing this particularly models works as compared to conventional neural networks (like convolutional neural networks)?

With the Gram Matrix itself being learnable and also patch-based, this CKDM architecture is reminiscent of an MLP-Mixer style architecture  which under certain circumstances (particularly reasonably large dataset) also performs competitively with Convolutional Networks. Would it be unreasonable to consider the CDKM itself as just yet another neural network architecture?

**Questions:**

Can the authors describe how this is more similar to a kernel method than a type of neural network since the Gram Matrices and mixup parameters are parameterized and trained with backpropagation?

**Limitations:**

Societal impact not applicable.

---

> ### Author Rebuttal · Authors · 2024-08-05
>
> Thank you for your review.
>
> > "In Table 1, the test-log likelihood improvements over previous work and traditional neural networks (without weight decay) is compelling as a result. But also surprised to see models with weight decay outperforming the other methods, if I understand correctly, CDKMs naturally provide uncertainty estimates for their predictions. Why would traditional models (with weight decay) be better at this?"
>
> It has been noted many times in the past that SGD leads to exceptionally good generalisation for ResNets [1], though the exact reason remains unclear. Although CDKMs are derived from Bayesian deep Gaussian processes, in order to retain representation learning, the precise infinite-limit taken means that we cannot expect all the benefits of fully-Bayesian methods to carry over when using DKMs. This point is discussed in more detail in previous DKM literature [2,3]. A better choice for uncertainty quantification would be a Deep Kernel Process or a Deep Gaussian Process. Nonetheless, uncertainty quantification was not our primary objective in this paper, but rather we sought to investigate representation learning in models other than neural networks, specifically kernel methods.
>
> > "Would it be unreasonable to consider the CDKM itself as just yet another neural network architecture"
>
> > "Can the authors describe how this is more similar to a kernel method than a type of neural network since the Gram Matrices and mixup parameters are parameterized and trained with backpropagation?"
>
> **A deep kernel machine (DKM) is not a neural network with a kernel component shoehorned in somewhere; a deep kernel machine never uses neural network features or weights anywhere in the architecture.**
> Instead, it works only with kernels, $\mathbf K$, or Gram matrices, $\mathbf G$ (which are very similar to kernels, they just have a slightly different use in the algorithm).
>
> To see quickly why a DKM is a kernel method and not a neural network, notice that the core "prediction" steps in Algorithm 1, just below the text "Predict train/test components of Gram matrix conditioned on inducing component" are
>
> $$\mathbf G_\text{ti} = \mathbf K_\text{ti} \mathbf K_\text{ii}^{-1} \mathbf G_{\text{ii}}$$
>
> $$\mathbf G_\text{tt} = \mathbf K_\text{ti} \mathbf K_\text{ii}^{-1} \mathbf G_\text{ii} \mathbf K_\text{ii}^{-1} \mathbf K_\text{ti}^T + \mathbf K_\text{tt} - \mathbf K_\text{ti} \mathbf K_\text{ii}^{-1} \mathbf K_\text{it}.$$
>
> These steps use the similarity of datapoints (as given by the kernel $\mathbf K$) to make predictions on the unseen datapoints, which is precisely what kernel methods like kernel ridge regression and Gaussian processes do. In other words, this is a function-space model. This is very different from neural networks / weight-space models, where we simply multiply our features by a set of weights.
>
> The confusion potentially lies in the fact that this is a kernel method with a **learned kernel**. Since the Gram matrices (along with other parameters of the variational approximate posterior) are learnable parameters, and we optimise them according to an objective function, it is convenient to use backpropagation for this purpose. Backpropagation is often associated with neural networks, but it is in fact used in a wide assortment of machine learning / statistical models. Also note that some other "learned kernel" approaches such as Deep Kernel Learning [4] actually use a neural network to map features before passing them to a traditional kernel function. Deep kernel machines don't use neural networks at any point, as discussed above. This therefore makes our empirical results quite significant.
>
>
> References:
>
> [1] Keskar, N.S. and Socher, R., 2017. Improving generalization performance by switching from adam to sgd. arXiv preprint arXiv:1712.07628.
>
> [2] Yang, A. X., Robeyns, M., Milsom, E., Anson, B., Schoots, N., and Aitchison, L. A theory of representation learning gives a deep generalisation of kernel methods. ICML, 2023.
>
> [3] Milsom, E., Anson, B., and Aitchison, L. Convolutional deep kernel machines, 2024, International Conference on Learning Representations (ICLR).
>
> [4] Wilson, A. G., Hu, Z., Salakhutdinov, R., and Xing, E. P. Deep kernel learning. In Artificial intelligence and statistics, pp. 370–378. PMLR, 2016.

---

> > ### Comment · Reviewer_YneB · 2024-08-09
> > **Thanks for the clarification**
> >
> > I have adjust my score based on the clarifications. To understand this better,
> >
> > What are the dimensions to the Gram Matrices? In Algorithm 1, how are the inducing points determined?

---

> > > ### Author Response · Authors · 2024-08-10
> > >
> > > Thank you for acknowledging our clarifications and increasing your score.
> > >
> > > When using the inducing point scheme, the learned inducing Gram matrices $\mathbf G_\text{ii}^\ell$ are size $P_\text{i}^\ell \times P_\text{i}^\ell$ where $P_\text{i}^\ell$ is the number of inducing points at layer $\ell$. This is because the Gram matrix represents the covariance between all pairs of points. Analogously, the block $\mathbf G_\text{ti}^\ell$ has size $P_\text{t} W_\ell H_\ell \times P^\ell_\text{i}$ where $P_\text{t}$ is the number of train/test points in the batch, and $W_\ell,H_\ell$ are the width / height of the image at layer $\ell$ (convolutions change the size of the image between layers). Finally, $\mathbf G_\text{tt}^\ell$ has shape $P_\text{t} W_\ell H_\ell \times P_\text{t}W_\ell H_\ell$, since it compares all pairs of test/train image pixels / patches. Note that, as in other convolutional kernel literature, it turns out that it is only necessary to store the diagonal of $\mathbf G_\text{tt}$, which would otherwise be very memory-intensive.
> > >
> > > All parameters in Algorithm 1 (including inducing inputs, inducing Gram matrices, and inducing outputs) are optimised using gradient-descent (Adam) on the DKM objective. The DKM objective is derived as an evidence lower-bound (ELBO), which motivates its use for optimising hyperparameters like the inducing points. Note that we initialise the inducing inputs as randomly sampled patches from the dataset, and initialise the inducing Gram matrices at $\mathbf G_\text{ii}^\ell = \mathbf K(\mathbf G^{\ell-1}_\text{ii})$, i.e. the NNGP, so the optimisation has a nice starting point. The inducing outputs are initialised with random mean and covariance, since the convolutional mixups make it difficult to meaningfully initialise these with data.

---

### Official Review · Reviewer_cbyd · 2024-07-10

**Soundness:** 3
**Presentation:** 3
**Contribution:** 2
**Rating:** 6
**Confidence:** 2

**Summary:**

The paper explores how to improve the convolution deep nuclear machines (DKMs)to improve their generalization ability, especially on the CIFAR-10 dataset. The authors introduced several modifications, especially random kernel regularization (SKR), which involves adding noise to the learned Gram matrix during training. The technique is inspired by discarding in neural networks and aims to reduce overfitting. Moreover, they use single-precision floating-point arithmetic to accelerate the training, thus allowing for more training cycles with a fixed computational budget.

**Strengths:**

Improving performance: The proposed modification greatly improves the test accuracy of DKMs on the CIFAR-10, from 92.7% to 94.5%.
New regularization techniques: Introducing stochastic kernel regularization is an innovative way to reduce overfitting in DKM.

**Weaknesses:**

Modifications involve complex changes to the DKM framework, which may be difficult to implement and understand for developers unfamiliar with these methods.

**Questions:**

How does the stability of the single precision calculation results ensure, and whether the article has additional requirements on the data set

**Limitations:**

yes

---

> ### Author Rebuttal · Authors · 2024-08-05
>
> Thank you for your review.
>
> > "Modifications involve complex changes to the DKM framework, which may be difficult to implement and understand for developers unfamiliar with these methods."
>
> We agree that the methods themselves are very novel and hence somewhat unfamiliar.  To address these issues, we have explained extensively the background (Section 2), methods (Section 3), and included a concrete algorithm with the changes from our work highlighted in red (Algorithm 1).
> Moreover, we have provided code with the paper, and we are currently in the processes of developing an easy-to-use PyTorch like library that incorporates these innovations.
>
> > "How does the stability of the single precision calculation results ensure, and whether the article has additional requirements on the data set"
>
> We include "number of failures" as a proxy for numerical stability in our ablation experiments. Section 4.2 provides further insight into the effects of our changes on the numerical stability of the model.
>
> We make no assumptions on the dataset.

---

### Official Review · Reviewer_44ig · 2024-07-12

**Soundness:** 2
**Presentation:** 2
**Contribution:** 2
**Rating:** 5
**Confidence:** 2

**Summary:**

This paper proposes a new method for Deep Kernel Machine, which achieves state-of-the-art results on kernel methods by using methods including regularization.

**Strengths:**

Compared to previous kernel methods, the proposed method achieves better results in CIFAR-10 test accuracy.

The use of stochastic regularization is straightforward and reasonable.

Some ablation studies are provided, including results with different hyper-parameters.

**Weaknesses:**

Though achieving state-of-the-art results for kernel methods, the proposed method is still relative time consuming and requires much resource.

According to the experiments, the proposed method might be sensitive to hyper-parameters and can lead to failure. Given the fact that it is already a time-consuming method, obtain a successfully trained model with good performance on new dataset might be very expensive.

CIFAR-10 is a relatively easy dataset, whether the proposed method works well on complicated cases are not investigated.

**Questions:**

Is there a comparison between results trained with double precision and single precision?

**Limitations:**

Yes

---

> ### Author Rebuttal · Authors · 2024-08-05
>
> Thanks for your review.
>
> > "Though achieving state-of-the-art results for kernel methods, the proposed method is still relative time consuming and requires much resource... sensitive to hyper-parameters"
>
> We agree that the method is time-consuming and sensitive to hyperparameters relative to SOTA neural network methods, and that these are important avenues for future work.  However, we do not believe that these are reasonable grounds for rejection, as our method is still SOTA for kernel methods by a considerable margin. Indeed, there are no kernel methods which are able to achieve close to this level of performance without facing similar issues.
>
> > "CIFAR-10 is a relatively easy dataset, whether the proposed method works well on complicated cases are not investigated."
>
> We have now additionally obtained performance-metrics on a different dataset, CIFAR-100. Our changes improve the test accuracy from 72.1\% in previous CDKM work [1] to 75.3\% in this work.
>
> > "Is there a comparison between results trained with double precision and single precision?"
>
> We do not provide direct comparisons between single and double precision due to the expense of running these experiments in double precision (modern GPUs are far better optimised for single precision), but one can compare our work to previous work on CDKMs [1] which utilised double precision, and observe that even in our ablations, there is no noticable degradation in predictive performance when using single precision arithmetic.
>
> References:
>
> [1] Milsom, E., Anson, B., and Aitchison, L. Convolutional deep kernel machines, 2024, International Conference on Learning Representations (ICLR).

---

> > ### Comment · Reviewer_44ig · 2024-08-13
> >
> > I would like to thank the authors for the clarification and additional results. Although I still have concerns on efficiency, I have increased the score.

---

> > > ### Author Response · Authors · 2024-08-13
> > >
> > > Thank you for considering our rebuttal and correspondingly increasing your score.

---

### Official Review · Reviewer_V8UE · 2024-07-28

**Soundness:** 4
**Presentation:** 3
**Contribution:** 3
**Rating:** 7
**Confidence:** 2

**Summary:**

The paper reports numerical results in which the authors have achieved state-of-the-art performances for an image classification task, namely the CIFAR-10 dataset, with a ''convolutional deep kernel machine''. The performance of this kernel-based model is close to the ones of state-of-the-art neural network architectures on the same dataset. To achieve this result, the authors introduce a ''stochastic kernel regularization'' procedure consisting of adding random noise to the parameter during training. They also use an approximation of the training objective which improves the numerical stability of training and allows to leverage the performances of modern GPUs to speed up the training procedure.

**Strengths:**

The paper is well-written, clearly introduces the ''convolutional deep kernel machine'' model it is studying, and clearly reports its methodology and its result. While I am not necessarily very familiar with the literature on deep kernel machines, the improvement in generalization with respect to previous works is significant and certainly contributes to closing a gap between kernel-based models and neural networks. Also the introduction of a stochastic regularization procedure for those kinds of kernel-based models seems like an interesting contribution.

**Weaknesses:**

In my opinion, the main weakness of the paper is to only provide results for the CIFAR10 dataset. It would be very enlightening to perform experiment for more involved image classification tasks, for example on Imagenet.

In addition, I think the paper sometimes lacks clarity in the way the training procedure is exposed. For example, in eq. 2 the kernel function $K$ is not defined. Also, the stochastic regularization term does not appear in eq. 2 while it is presented as the objective function used during training. To address this issue, the authors could add a precise description of the training algorithm (in the same spirit as algorithm 1 but in my understanding algorithm 1 describes the algorithm for inference on the training/test data, and the last line about training is quite elusive).

**Questions:**

- Did you try to perform experiments on other datasets such as Imagenet ? Should we expect deep convolutional kernel machines with stochastic kernel regularization to have performances similar to ResNets or would they perform way worse ?

- One of the encountered problem during training is that the condition number of the Gram matrices $G^\ell_{ii}$ tends to worsen over time. Isn't that behavior to be expected when the number of inducing points is larger than the number of classes ? In my understanding, if the features are sufficiently expressive, the rank of the Gram matrices should be equal to the number of classes. Do you agree with this intuition ? Did you try to compute the rank (or eigenvalue distribution) of the Gram matrices ?

- It is written that ''we expect the Gram representations to be close to those
  of the NNGP'', could you provide a justification of this statement ?

The rest of my questions are concerned with some incoherences which might simply be typos but hinder my comprehension of the paper. Please correct me if I simply misunderstood those equations.

- At the end of section 4.2 you write several times $G^{-1} K G^{-1}$, shouldn't it be $G^{-1} K$ ?

- In eq. 9b you write $K_{features}(F^\ell)$, shouldn't it be $K_{features}(F^{\ell-1})$ ? Also in this equation it is not really clear for me what the function $K_{features}$ is.

- In eq. 14 shouldn't the divergence term be $D_{KL}(N(0, G^\ell)||N(0, K(G^{\ell-1}))$ ?

- In eq. 15b shouldn't it be $K_{features}(F^{\ell-1})$ ?

- In eq. 16 shouldn't it be $\mathbb{E} \left[ F^\ell_{ir,\mu} F^\ell_{js,\mu'} | H^\ell \right]$ ?

- In eq. 21 shouldnt' it be $\Gamma(K(G^{\ell-1}))$ ?

- In eq. 25 it should be $F^{\ell-1}$.

- I do not understand eq. 29. The r.h.s. uses $\Gamma^\ell_{tt}$ which, if I understand correctly, is defined using the whole feature matrix $F^\ell$ whose probability density we are trying to compute. Shouldn't it be $\Gamma^{\ell-1}$ in eq. 29 ?

**Limitations:**

Limitations of this work are discussed in the dedicated section. On top of that, I think that an important limitation is to only provide results for the CIFAR10 dataset. It could be discussed if one should expect convolutional deep kernel machines to have performances similar to the ones of neural networks for more involved image classification problems.

---

> ### Author Rebuttal · Authors · 2024-08-05
>
> Thank you for your detailed review.
>
> > "...main weakness of the paper is to only provide results for the CIFAR10 dataset"
>
> We have managed to secure enough compute resources to also benchmark our method on the CIFAR-100 dataset. We improve the CIFAR-100 test accuracy from 72.1\% in earlier CDKM literature [1] to 75.3\% with our methods.
>
> > "I think the paper sometimes lacks clarity in the way the training procedure is exposed ... To address this issue, the authors could add a precise description of the training algorithm"
>
> Thank you for this feedback. Condensing complex background material down so there is still space for novel contributions can be tricky, so hopefully incorporating your suggestions in our working draft will improve readability.
>
> To address some specific questions:
>
> > "the stochastic regularization term does not appear in eq. 2 while it is presented as the objective function used during training"
>
> The objective function consists of 2 terms: the log-likelihood term, which encourages good performance on the training data, and the sum of KL divergences, which encourages the representation of each layer to be similar to the previous layer (a form of regularisation). We use the samples from "stochastic kernel regularisation" (SKR) during the forward pass, as described in the algorithm, which affect the log-likelihood term, but do not modify the KL divergence terms, meaning the KL terms still act directly on the original parameters $\mathbf G^\ell_\text{ii}$, i.e. the means of the SKR samples. This is because only the log-likelihood term can cause overfitting to data, so there is no good reason to use the random samples in the regularising terms.
>
> > "the last line about training is quite elusive"
>
> We have updated this to be easier to understand. Training is similar to any other pytorch model, in that we input data into our model, compute a loss function (in our case a rather complex DKM objective function) and then update the parameters of the model by backpropagation.
>
> > "One of the encountered problem during training is that the condition number of the Gram matrices tends to worsen over time. Isn't that behavior to be expected when the number of inducing points is larger than the number of classes?"
>
> Agreed, this is expected, at least in later layers.
> However, we found empirically that good performance required a number of inducing points far beyond the number of classes, so we only asked the simpler question "how can we encourage these Gram matrices to be better conditioned?".
>
> > "It is written that "we expect the Gram representations to be close to those of the NNGP", could you provide a justification of this statement?"
>
> This is based on the intuition that we regularise our model to be close to the NNGP via the KL divergence terms in the objective (Eq. 2), combined with the fact that we initialise at the NNGP (i.e. set $\mathbf G^\ell = \mathbf K^{\ell-1}$).
>
> >"At the end of section 4.2..."
>
> This is a confusion between matrix multiplication and function composition; we will simplify the notation here to make it clearer.
>
> Regarding typos in the appendix (we especially appreciate the reviewer's efforts in reading the appendix in addition to the main text):
>
> >"In eq. 9b..."
>
> Fixed. $K_\text{features}$ is just a traditional kernel like arccos or sqexp, we have now added a clarifying comment.
>
> >"In eq. 14 shouldn't the divergence term..."
>
> >"In eq. 15b shouldn't it..."
>
> >"In eq. 16..."
>
> >"In eq. 21..."
>
> >"In eq. 25..."
>
> All fixed. Many thanks.
>
> >"I do not understand eq. 29..."
>
> This is again a mixup between $\ell$ and $\ell -1$. We have thoroughly proofread the appendix and made sure these issues are now fixed / consistent.
>
> References:
>
> [1] Milsom, E., Anson, B., and Aitchison, L. Convolutional deep kernel machines, 2024, International Conference on Learning Representations (ICLR).

---

> > ### Comment · Reviewer_V8UE · 2024-08-14
> >
> > I would like to thank the authors for their answers and clarifications and will keep my score as it is.
> >
> > Regarding the paper, I still suggest clarifying the training procedure. Also if numerical results are available for CIFAR100 it would be beneficial to mention it in the paper.

---

### Author Rebuttal · Authors · 2024-08-05

We thank all the reviewers for their helpful comments. All reviewers recognised that our work presents state-of-the-art results for kernel methods on CIFAR-10.

> "the improvement in generalization with respect to previous works is significant and certainly contributes to closing a gap between kernel-based models and neural networks" - Reviewer V8UE

> "Compared to previous kernel methods, the proposed method achieves better results in CIFAR-10 test accuracy." - Reviewer 44ig

> "The proposed modification greatly improves the test accuracy of DKMs on the CIFAR-10, from 92.7\% to 94.5\%." - Reviewer cybd

> "They achieve 94.52\% test accuracy on CIFAR-10, which is a significant improvement over the previous best kernel method result of 92.69\% and comes close to matching comparable neural network performance" - YneB

This was made possible through our novel methods to the CDKM model:

> "Introducing stochastic kernel regularization is an innovative way to reduce overfitting in DKM." - Reviewer cybd

> "the introduction of a stochastic regularization procedure for those kinds of kernel-based models seems like an interesting contribution" - Reviewer V8UE

> "their improvements in numerical stability allow for the use of TF32 arithmetic, making training about 5 times faster than previous implementations" - Reviewer YneB

> "They introduce a novel regularization technique where the learned inducing Gram matrices are randomly sampled from a Wishart distribution during training." - Reviewer YneB


Some reviewers would have liked to see more empirical evaluation of our methods. To address this, we managed to secure enough compute to evaluate our method on an additional dataset, CIFAR-100. With our new techniques, CDKMs achieve 75.31\% test accuracy on CIFAR-100, which represents a \~3\% improvement over the previous best result from the DKM literature, which was 72.05\% [1].

We have also made some minor changes to our working draft in order to improve readability, following the helpful suggestions of the reviewers, but no serious issues about presentation were raised:

> "The paper is well-written, clearly introduces the ''convolutional deep kernel machine'' model it is studying, and clearly reports its methodology and its result." - Reviewer V8UE

References:

[1] Milsom, E., Anson, B., and Aitchison, L. Convolutional deep kernel machines, 2024, International Conference on Learning Representations (ICLR).

---

### Decision · Program_Chairs · 2024-09-25

**Decision:**

Accept (poster)

**Comment:**

Deep Kernel Machines are an interesting theoretical tools: this paper aims at boosting the performance of those theoretical models, by introducing several regularization (notably, noise)

The main concern of the reviewers is that this paper should have considered also challenging datasets like imagenet. Yet,  given the significant improvement on cifar10, both from the accuracy perspective and speed, I recommend accepting this paper.